# Remote, Whole-Body Interval Training Improves Muscular Endurance and Cardiac Autonomic Control in Young Adults

**DOI:** 10.3390/ijerph192113897

**Published:** 2022-10-26

**Authors:** Patricia Concepción García-Suárez, Ermilo Canton-Martínez, Iván Rentería, Barbara Moura Antunes, Juan Pablo Machado-Parra, Jorge Alberto Aburto-Corona, Luis Mario Gómez-Miranda, Alberto Jiménez-Maldonado

**Affiliations:** 1Facultad de Deportes Campus Ensenada, Universidad Autónoma de Baja California, Ensenada 22800, Mexico; 2Department of Health, Sport and Exercise Sciences, University of Kansas, Lawrence, KS 66045, USA; 3Facultad de Deportes Campus Tijuana, Universidad Autónoma de Baja California, Tijuana 22390, Mexico

**Keywords:** high-intensity interval training, COVID-19 pandemic, heart rate variability

## Abstract

High-intensity interval training (HIIT) is an exercise modality acknowledged to maintain physical fitness with more engagement in an active lifestyle compared with other traditional exercise models. Nevertheless, its effects on cardiac control and physical performance in an online-guided setting are not yet clarified. The present work assessed physical fitness and heart rate variability (HRV) before and after an online, home-based HIIT program in college-age students while pandemic lockdowns were in effect. Twenty university students (age: 21.9 ± 2.4 years.) that were solely enrolled in online classes were distributed into three groups: control—CON-(*n* = 6), 14 min of HIIT—HIIT-14-(*n* = 8), and 21 min of HIIT—HIIT-21-(*n* = 6). A maximal push-up test was employed to assess muscular endurance and performance, and resting HRV signals were collected with wireless heart rate monitors and were processed in Kubios HRV Std. (Kubios Oy, Finland). There was an increase in total push-up capacity compared to CON (*p* < 0.05 HIIT-21 vs. CON; *p* < 0.001 HIIT-14 vs. CON) after 8 weeks. A significant interaction was observed in high-frequency and low-frequency spectra ratios after the HIIT-21 intervention (*p* < 0.05). The current work demonstrated that either short- or mid-volume online, whole-body HIIT improves muscle strength, whereas mid-volume HIIT (HIIT-21) was the only intervention that developed a sympathovagal adaptation. This study showed promising results on muscular endurance and cardiac autonomic modulation through whole-body HIIT practice at home.

## 1. Introduction

The pandemic of the 21st century increased sedentary behavior (SB) in most of the population, causing people to become prone to acquiring morbid conditions associated with a lack of physical activity (PA) [1,2,3]. This negative effect was magnified in all ages, countries, and sexes [3,4].

The negative consequences of pandemic SB were akin to non-pandemic SB, such as adipose tissue gain, muscle mass loss combined with lower physical aptitude, and a rise in low-grade inflammation, and these disorders increased the risk for the development of metabolic, mental, and cardiovascular diseases (CVD) [4,5,6]. The latter, combined with the current morbid state and lifestyle habits of developed countries and their adjacent countries, induced an increase in case–fatality rates for cardiovascular events and CVD during and after the COVID-19 pandemic [6,7,8,9]. Reports also acknowledge that these mortality rates were related to postponed CVD procedures and reduced medical services within the first two years of the pandemic [7,8]. Although the provision of medical services is crucial for the treatment of CVD by pharmacological therapies, non-pharmacological strategies such as physical exercise (PE) are considered to be a cost-effective strategy to prevent and reduce CVD [10].

Therefore, international health authorities advise keeping regular physical conditioning during health emergency lockdowns [1,11]. The recommendations include a broad range of home-based training without the expense of costly equipment, such as aerobic conditioning as well as moderate-to-low intensity and high-volume adaptive resistance training [11], following the international guidelines for session volume and frequency [12], with an average of 150 min per week for the adult population [13,14]. Even though these recommendations were massively promoted to the general public, novel barriers such as home-office duties; increased domestic chores; and remote homeschooling for children, adolescents, and college-age students impeded PA commitment during the COVID-19 lockdown [1,15]. These barriers also contributed to the previously reported lack of time and motivation to engage in an exercise regime in pre-pandemic times [16]. 

In this sense, high-intensity interval training (HIIT) is an exercise model with brief, high-intensity outbursts and short-to-medium resting periods [17,18]. This exercise model showed a broad range of versatility in laboratory, field, and at-home settings due to its little-to-null equipment utilization in a reduced amount of time (i.e., ~< 25 min.) [19,20,21], addressing a practical solution for compelling the PE guidelines while keeping a responsible, at-home lockdown. Nevertheless, surveillance and guidance from physical activity and sports professionals were a factor for successful, remote/home-based PE interventions. The solution proposed by research teams around the globe is centered around the utilization of wireless technology with non-invasive physiological markers [22,23,24].

Heart rate variability (HRV) is a compelled-signal characterization of the autonomic modulation of the heart by the vagal and sympathetic nerves [25,26]. The signal processing in time and frequency series of HRV can elucidate parasympathetic activity or a combination of both parasympathetic and sympathetic behavior in clinical conditions such as CVD risk [27,28,29], inflammation [30,31], stress, and emotional regulation [32,33,34]. Studies indicate that parasympathetic HRV indexes are predictors of the cardiorespiratory fitness of young adults [35,36] after VO_2max_ and HR [37]. While this physiological marker is a good indicator of cardiovascular health, muscle strength, and endurance, there are other elements necessary to survey physical fitness and performance [38,39,40]. Both variables have shown significant improvements in a controlled-environment manner [38,41,42]. However, the effects of remote, home-based HIIT on both cardiac and muscular endurance, while keeping the safety measures of the COVID-19 lockdown, is incipient.

Therefore, the aim of the present work was to assess the effects of equipment-free, online, guided HIIT on heart rate variability markers and physical endurance in isolated college-age adults during the COVID-19 pandemic. Opposite previous online-, tele-, or mobile-health interventions with designs centered towards volitional-based enclosure, we aimed to assess the physiological responses under a non-elective lockdown. The secondary aims of the study revolved around the dose–response of the HIIT’s session volume and gradual HRV and muscle endurance adaptations. The novelty of this work is that it aimed to address the magnitude of whole-body, equipment-free HIIT through cardiac markers and muscle endurance in a remote, unsupervised manner during the COVID-19 lockdown.

## 2. Materials and Methods

### 2.1. Participants

Twenty-seven college-age adults that were enrolled in the Facultad de Deportes at Universidad Autónoma de Baja California (age: 20.8 ± 0.9 years; BMI: 24.5 ± 4.8 kg·m^−2^) were recruited for this study. The inclusion criteria were that volunteers had no physical injuries or limitations that would prevent them from performing high-intensity physical activity. The researchers only included those volunteers who proceeded to sign the informed-consent letter and felt comfortable participating in the study despite the prolonged-lockdown side effects. The exclusion criterion was any participant or family member living at home testing positive for COVID-19 throughout the study. Using simple randomization with an Excel spreadsheet, participants were randomly assigned to either 14-min HIIT (HITT-14), 21-min HIIT (HIIT-21), or control (CON). Seven participants withdrew from the study due to personal reasons, leaving a final sample of twenty subjects (HIIT-14 *n* = 8, HIIT-21 *n* = 6, and CON *n* = 6).

### 2.2. Study Design

Sample size estimation was determined a priori with G*Power ver. 3.1.9.4. (Universität Kiel, Germany). For design selection, repeated measures between–within interaction with the F-test family was chosen, with 3 groups and 2 repeated measures a moderate effect size of f = 0.4, α = 0.05, and power (1 − ß) of 0.8. The minimum sample-size output was 21. Due to the dropout of participants before the completion of the study (*n* = 7), we achieved a marginally required sample of 20 individuals for the final analysis. 

For recruitment, the researchers sent electronic invitations by e-mail to students in their second year of their bachelor’s degree from the Facultad de Deportes Campus Ensenada at Universidad Autónoma de Baja California. In the electronic message, the authors briefly explained the aims of the study. The interested people attended a virtual meeting (Google Meet) where the researchers explained in detail the aim and procedures of the study. After this, the volunteers read and signed a written informed consent set up in Google Forms. After that, strictly following the guidelines of the World Health Organization (WHO) to prevent the spread of COVID-19, the participants assisted at the Laboratorio de Fisiología Aplicada al Ejercicio Físico (LaFAEF) in the Facultad de Deportes Campus Ensenada, where the academic staff gave a wireless heart rate monitor for data collection on the Elite HRV^®^ app at rest and during the exercise sessions. Additionally, during this visit, the students’ body weight and height were measured with a digital floor scale (RICE LAKE, Rice Lake, WI, USA) and a portable stadiometer (Biospace, Seoul, Korea), respectively. 

In regards to exercise intervention, a videotape was filmed by qualified fitness trainers to demonstrate the HIIT sessions to the participants in the exercise groups, whereas the CON group was asked to perform their usual routines according to their lifestyle during the lockdown. With the aim to subjectively measure the physical activity levels of each participant, they completed an electronic version of the short-form International Physical Activity Questionnaire (IPAQ-SF). The IPAQ-SF consists of seven questions which recall the previous week’s physical activity and asks them about the number of days and the amount of time spent walking, sitting, or participating in moderate- and vigorous-intensity activities. Additionally, the Physical Activity Readiness Questionnaire (PAR-Q) was applied in order to assess the participants’ health status for performing highly demanding exercises via Google Forms. The program was carried out from April to June 2021. All procedures were approved by the Research Ethics Committee of Facultad de Medicina y Psicología Campus Tijuana, and the protocol was registered under the code 889/2020-2.

### 2.3. HIIT

The HIIT program lasted a total of 8 weeks, performing the HIIT circuit 3 times a week for the HIIT-14 and HIIT-21 groups. The exercise sessions consisted of a warm-up of 2 min encompassing stretching exercises. Immediately after warm-up, participants started a modified version of the program described by Schleppenbach et al. [20]. Concretely, whole-body exercises, including jumping jacks, high knees, burpees, push-ups, sit-ups, jump rope, and line jumps, were executed with as many reps as possible for 30 s, followed by 30 s passive recovery between exercises. The completion of 2 sets (14 min) was applied to the HIIT-14 group, while the HIIT-21 group was asked to carry out 3 sets (21 min). Throughout all the exercise interventions, the participants wore a heart rate monitor.

### 2.4. HRV

Before HRV collection, all participants were asked to avoid drinking caffeine and alcohol and to refrain from strenuous physical exercise for at least 24 h before evaluation. Baseline measures were recorded in the morning (8:00–10:00 a.m.) via a Bluetooth heart rate monitor (Polar H10, Polar, Inc. Kempele, Finland). The participants were instructed to remain in a supine position for 10 min before the beat-to-beat intervals (R–R) were registered using the Elite HRV^®^ app (Elite HRV LLC, Asheville, NC, USA, and Release 4.0.2, 2018) in their mobile phones. Likewise, every HIIT session was logged and recorded in the app. Throughout the exercise program, a staff member from the research team sent a weekly reminder to the participants to share their raw R-R signals (export option in the mobile app) in a shared Google Drive folder. Raw R-R signals were analyzed with Kubios Heart Rate Variability standard version software (Kubios HRV 3.4.2, Oy, Kuopio, Finland). The beat correction was set at very low to avoid possible noise data associated with involuntary movements, and the last 5 min of each recording was framed for the final analysis, following the suggested literature [43]. Time–domain parameters of mean heart rate (HR), mean beat-to-beat intervals (IBI), standard deviation of beat-to-beat intervals (SDNN), root mean square of the successive differences (RMSSD), and direct and relative successive beats with >50 ms of difference (NN50 and pNN50) were estimated. Finally, the frequency domains for very-low-frequency (VLF); low-frequency (LF); and high-frequency (HF) in normal and raw units (n.u. and ms^2^), LF/HF ratio, and raw total power were collected. Frequency cut-off values were set at <0.049 Hz for VLF, 0.05–0.15 Hz. for LF, and 0.16–0.4 Hz. for HF with the Fast Fourier Transformation method (FFT).

### 2.5. Maximal Push-Up Test

In order to keep the main purpose of the study, the research team selected a single test to keep the least intrusive setting intervention possible. A simple and versatile way to summarize physical fitness was with maximal push-ups. This assessment is strongly related to cardiovascular markers and physical performance [38,40]. Muscular endurance was assessed with the push-up test; it was administrated following the recommended guidelines [39,44]. The participants were asked to carry out push-ups until fatigued. The push-up test was applied every 2 weeks online (Google meet), under staff research supervision.

### 2.6. Statistical Analysis

Two-way mixed ANOVA design (Time (pre vs. post) and Treatment (CON vs. HIIT-14 vs. HIIT-21)) was applied to distribute the data and to assess the magnitude of change in HRV variables and muscle endurance. Levene’s test for equality of variances was applied to all variables, resulting in a violation of equality of variance in the LF/HF ratio, and Welch’s adjustment was implemented on the 2-way ANOVA model. Statistical significance was set at *p* ≤ 0.05, and data were reported as mean ± SD for main and interaction effects. Additionally, 95% confidence intervals (95% CI) and eta-squared (ɳ^2^) effect size (0.01 trivial, 0.06 medium, and ≥0.14 large) were computed for any significant effect on the post hoc analysis for both main and interaction effects [45]. All statistical analyses were performed on R software, using the integrated development environment RStudio team for data management and analysis. The packages *lme4*, *sjstats*, *psych*, and *ggplot2* were employed for the inferential, descriptive, and figure designs, respectively.

## 3. Results

### 3.1. Exercise Program

Demographic variables are addressed in Table 1. The total exercise-log registry was 91.4% combined for the HIIT-14 and HIIT-21 groups from a total of 40 and 24 sessions each. Maximum HR intensity reached was 157 ± 19 bpm, with an average HR recovery of 120 ± 19 bpm for HIIT-14 during the training sessions. For group HIIT-21, the maximum HR was an average of 192 ± 23 bpm with an average recovery period of 125 ± 28 bpm. This gave the grand average HR per session of 154 ± 18 bpm (78.1% HRmax) and 149 ± 13 bpm (76.2% HRmax) for each group, respectively. 

### 3.2. Muscular Endurance

For push-up capacity, there was a significant interaction for the HIIT-14 group, with no other significant change across the 8 weeks of isolation in the main effects of time (wk8 vs. baseline, *p*: 0.688, 95% CIs: −4.59–6.93, ɳ^2^: 0.25) or group (HIIT-14 vs. CON, *p*: 0.819, HIIT-21 vs. CON, *p*: 0.450, 95% CIs: −13.66–10.83 and −18.09–8.09, ɳ^2^: 0.11). The significant interaction of the HIIT-14 showed a push-up improvement after the sixth week when compared to baseline with a moderate effect (baseline: 24.75 ± 5.7, wk6: 35.5 ± 8.28, wk8: 36.12 ± 7.3, *p*: 0.009, ɳ^2^: 0.15) (Figure 1), showing a marginal trend to increase upper-body strength during the intervention. Regarding HIIT-21, a marginal interaction was found by the end of wk8 (baseline: 21.16 ± 11.32 reps vs. wk8 29.83 ± 16.82 reps, *p*: 0.071, 95% CIs: −0.64–15.64). 

### 3.3. Heart Rate Variability

Full HRV analyses are displayed in Table 2. There were no modifications for most of the HRV variables throughout the intervention (*p* > 0.05). Nevertheless, an interaction effect was found for IBI (*p*: 0.029), VLF (*p*: 0.021), and LF/HF (*p*: 0.05), and spectral adaptations from HIIT-21 were observed.

## 4. Discussion

The purpose of our study was to assess the impact of an unsupervised, remote HIIT program on HRV and muscle endurance during a prolonged lockdown (~8 weeks). Our exercise groups showed an augmentation of muscle endurance when compared to the control, while the HIIT-21 was the only group with HRV alterations on IBIs, VLF and LF/HF after 8 weeks of unsupervised training during a lockdown. For the IBI increments, even if not yet reflected by a significant HR reduction in our HIIT-21 group, the marginal *p*-value in combination with a big effect size (Table 2) showed a behavioral trend to reduce HR. This phenomenon was also reported by an earlier study by Marshall et al., where the researchers applied unsupervised aerobic training in middle-aged adults 5 times a week for 8 weeks [46]. The authors concluded that constant aerobic exercise was not enough stimulus to unveil cardiac functional gains in the interceded sample. Likewise, other reports denoted that 12 weeks of intervention induces HR reduction in age-matched subjects, such as in our sample [41,47]. Thus, a more-prolonged, unsupervised program could have helped to denote a major significance in HR reduction. 

Cardiac adaptations, including the decline in baseline HR following exercise programs, have been widely documented among different populations and conditions [48,49,50]. However, we cannot address whether the IBI increments or other time-series adaptations are exclusive to vagal modulation. This is due to the fact that all time series derive from HR subcalculations/indexes, which reduce beating at baseline and can be mediated either by parasympathetic tone activity or sympathetic withdrawal [25]. The time-series measures are not enough of an indication to elucidate which of them modified the IBI increase [25,51]. 

Regarding the spectral series adaptations, a VLF and an LF/HF ratio were observed for the HIIT-21 group, and no HRV indicators were observed for HIIT-14 after 8 weeks of training. The VLF modification was an unusual marker whose physiological interpretation has not been clarified yet [43]. The VLF was necessary for the n.u. characterization of the remaining two main spectral components (i.e., LF and HF) and the subtraction of the total power signal [43]. Therefore, the VLF increase after 21 min in the HIIT group played a key role in the marginal n.u. modification of the HF and LF components; that is, there was no significant *p*-value, but there was a large effect size. Similarly, the VLF change contributed to the reduction in the LF/HF ratio in the HIIT-21 participants, indicating an improvement in cardiac sympathovagal modulation [25,52]. A meta-analysis conducted by Shenoy et al. remarked on the HRV adaptations in HIIT protocols [52], and our findings, with respect to the LF/HF ratio reductions, agreed with the latter report. Nevertheless, the RMSSD and HF were not modified in either HIIT intervention. Unlike the other reports, our study utilized 14 and 21 min of total volume per session, a minor time employed in comparison to other authors who applied larger volume durations in their programs (~25–60 min) [41,52]. The null effects of HIIT-14 on HRV variables were compatible with the experiments conducted by Soltani et al., where the researchers determined that HRV adaptations are intensity- and volume-dependent [41]. Thus, the home-based HIIT volume was dependent on time in our groups.

In regard to the push-up data, both exercise groups improved in the fitness category after the intervention. Specifically, the HIIT-14 group started at good fitness and finished in the excellent fitness category, whereas the HIIT-21 group at baseline was classified as being in the acceptable fitness category and, after the intervention, achieved the very-good fitness category [44]. Therefore, it is possible to consider homemade interval training as an efficient intervention to improve muscular endurance in the university, as there was a remarkable improvement by the HIIT-14 group but not as much for the HIIT-21 group. Reiteratively, this seems to be volume-dependent for a chronic exercise program (14 min vs. 21 min). These data concord with the reports for remote programs where unsupervised home-based exercise aids in improving physical conditioning, similar to that of laboratory-based training [53]. Moreover, our remote, 14-min circuit training seems to not just prevent the loss of muscular-endurance capacity but to improve it. These data are contradictive to other studies where the population was first introduced to a supervised program and later allowed “free-will” training at home. Fenell et al. improved the push-up capacity after 5 months of group training, just to return to baseline values after a 5-week unsupervised training period [38]. 

The exercise bouts’ duration and session volume during unsupervised programs are elements that were conjointly studied over 30 years ago [54,55]. However, the emerging circumstances of the pandemic oblige health professionals to bring novel information for better physical health monitoring and conditioning with remote technology and inexpensive equipment [3,22,56]. Our study showed that different circuit repetitions are dose-dependent for cardiac and performance modifications. One possible explanation is that the 14 min HIIT might not be enough of a stimulus to make HRV adaptations appear. On the other hand, 21 min of HIIT might have altered a greater effort to enhance muscle endurance due to the accumulative lactate and peripheral fatigue that whole-body exercises can cause in a longer run [57]. Ojeda et al. provided insight into how whole-body exercises, based on intervallic burpees, create less exertion in an ~6 min protocol than similar intensity-matched exercises [57]. 

This study has some limitations: (1) the optimal sample size was affected by a 26% reduction in the original sample; thus, cautionary conclusions from these effects might be taken into account, such as its contribution to the borderline *p*-values on some HRV markers. Furthermore, HRV is related to mental distress [32]. (2) Notwithstanding the fact that we did not evaluate mental stress or perceived social isolation in our sample, the COVID-19 enclosure caused a spike in mental conditions in most of the population [33,58]. Hence, the secure lockdown and isolation might have contributed to weakened HRV adaptations in our intervention with young adults [33]. Further studies controlling for mental state and wellness can aid in finding a relationship between homemade PE plans and HRV/fitness changes.

## 5. Conclusions

Overall, the current study showed that virtual, whole-body interval training is an intervention that improves muscular endurance in the young-adult population under mid-term lockdown. Moreover, the same intervention seemed to not induce physiological stress (regarding the HRV data) in a relatively small sample. These results can be useful for health professionals, mainly for making recommendations about performing physical activity during social restrictions such as those recently experienced by society. 

Therefore, future large-scale or long-term studies on this topic can be accomplished due to relatively easy data acquisition, and no specific care is required for the participants. 

## Figures and Tables

**Figure 1 ijerph-19-13897-f001:**
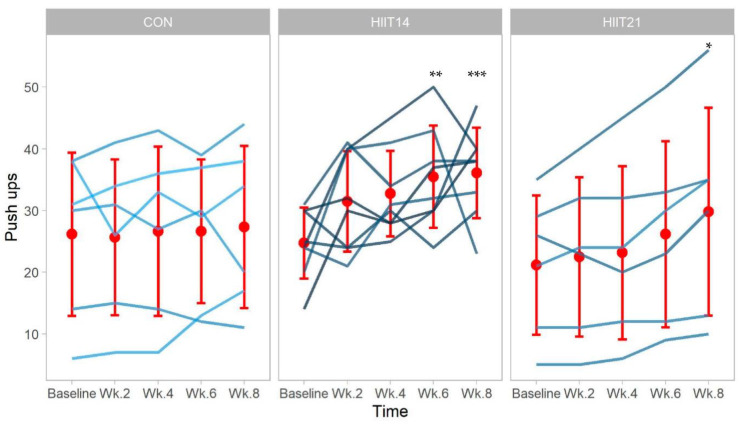
Individual push-ups during the 8 weeks of confinement, monitoring for CON (control), HIIT-14 (14), and HIIT-21 (21). *** (*p* < 0.001 vs. baseline), ** (*p* < 0.01 vs. baseline), and * (*p* < 0.05 vs. baseline). Red dots: Mean values of every week; red bracket lines: standard deviations of the time measure.

**Table 1 ijerph-19-13897-t001:** Demographic characterization of the participants.

Characteristics	CON (*n* = 6)	HIIT-14 (*n* = 8)	HIIT-21 (*n* = 6)
Age (years) *	23.66 ± 2.6	20.85 ± 1.7	22.50 ± 2.5
BMI (kg·m^−2^) *	21.74 ± 2.7	30.87 ± 14.7	26.42 ± 6.8
Physical activity (METs·min·wk^−1^) ^1,^*	1416 ± 1389	1372 ± 834	2205 ± 1446
Male to female ratio	5/1	5/3	5/1

^1^ Estimated energy expenditure according to the IPAQ-SF. * No significant difference between the groups’ (*p* > 0.05) one-way ANOVA.

**Table 2 ijerph-19-13897-t002:** HRV variables.

Group	CON		HICT-14		HICT-21		*p* Values	ɳ^2^
Time/Variable	Pre	Post	Pre	Post	Pre	Post	Time Treatment Interaction	Effect Size (95% CIs)
HR (bpm)	68 ± 15	69 ± 6	63 ± 11	63 ± 13	83 ± 33	64 ± 14	0.886 0.523 0.074	0.21 (−2.06–42.73)
SDHR (bpm)	5.38 ± 2.1	4.27 ± 1.4	4.88 ± 1.9	4.43 ± 1.8	7.11 ± 4.0	4.44 ± 1.2	0.315 0.898 0.650	0.13 (−2.25–3.56)
IBI (ms)	921 ± 188.8	877 ± 80.2	979 ± 194.7	990 ± 206.7	**789 ± 202.9**	**981 ± 221.7 ***	0.551 0.278 **0.029 ***	**0.19** (−378.25–15.68)
SDNN (ms)	67.94 ± 12.8	54.53 ± 21.1	68.62 ± 20.8	66.00 ± 22.0	62.99 ± 28.5	73.38 ± 43.0	0.166 0.419 0.085	0.128 (−51.05–3.45)
RMSSD (ms)	55.35 ± 17.5	48.73 ± 19.4	71.85 ± 24.4	69.79 ± 30.3	59.48 ± 25.4	81.68 ± 52.0	0.617 0.205 0.794	0.14 (−39.84–30.71)
pNN50 (%)	31.38 ± 18.6	22.73 ± 19.2	39.75 ± 13.7	38.78 ± 16.7	35.67 ± 21.5	41.67 ± 24.3	0.330 0.126 0.511	0.08 (−31.24–15.86)
VLF (ms^2^)	351.89 ± 309.65	101.75 ± 57.7	262.54 ± 257.2	338.85 ± 213.4	**101.32 ± 89.2**	**384.47 ± 539.9 ***	0.117 0.134 **0.021**	**0.26** (−980–−86.52)
LF (ms^2^)	3097.72 ± 1956.6	2029.85 ± 1314.5	2030.98 ± 1309.3	2153.78 ± 1460.3	3034.96 ± 3910.17	3049.26 ± 4082.2	0.216 0.928 0.372	0.07 (−351767–−1353.35)
HF (ms^2^)	1815.67 ± 1469.4	1480.04 ± 1637.2	2192.97 ± 1348.8	1935.80 ± 2080.4	962.15 ± 1243.1	2394.34 ± 2386.9	0.680 0.632 0.131	0.16 (−4088.42–552.77)
LF (n.u.)	62.33 ± 20.9	60.05 ± 20.5	48.54 ± 10.1	56.31 ± 15.1	68.68 ± 20.4	51.70 ± 12.9	0.681 0.395 **0.068**	**0.34** (9.91–39.59)
HF (n.u.)	37.65 ± 20.9	39.92 ± 20.5	51.43 ± 10.1	43.65 ± 15.0	31.24 ± 20.44	48.24 ± 12.9	0.683 0.396 **0.068**	**0.34** (−39.61–−9.94)
LF/HF	2.37 ± 1.7	2.42 ± 2.2	1.00 ± 0.3	1.70 ± 1.4	**5.51 ± 7.1 ^§^**	**1.22 ± 0.72 ***	0.973 0.667 **0.05**	**0.12** (0.07–8.61)
Total Power (ms^2^)	5265.55 ± 2127.7	3612.21 ± 2274.1	4487.46 ± 2721.1	4429.49 ± 3560.4	5075.99 ± 4332.4	5828.81 ± 6082.8	0.253 0.686 0.239	0.08 (−6495.34–1683.00)

Abbreviations: IBI—mean inter-beat interval length, HR—mean heart rate, SDNN—standard deviation of inter-beat intervals, RMSSD—root mean square of the successive differences, pNN50—relative successive beats with > 50 ms of difference, VLF—very-low-frequency domain, LF—low-frequency domain, HF—high-frequency domain, LF/HF—low-to high-frequency domain ratio, ms—milliseconds, bpm—beats per minute, n.u.—normal units. ^§^ Adjustment after Welch’s correction for unequal variances. **Bold characters***: Significance *p* < 0.05, 2 × 3 Mixed ANOVA.

## Data Availability

The data presented in this study are available upon request from the corresponding author. The data are not publicly available due to privacy issues.

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
