# Peer review of "Remote, Whole-Body Interval Training Improves Muscular Endurance and Cardiac Autonomic Control in Young Adults"

_ijerph, 2022, doi:10.3390/ijerph192113897_

Round 1
Reviewer 1 Report
The presented article "Online Whole-Body Interval Training Improves Muscle Performance and Spectral Cardiac Autonomic Control in Young Adults " clearly showed that virtual whole-body interval training was an intervention that improved the muscular endurance in young-adults population.
The experimental design is straightforward and the manuscript is pretty well written, although some English proofing would be beneficial. For example:
- lane 4 – there shouldn’t be a period in the headlines
- lane 40 – there shouldn’t be an “s” in adipose tissue gain,
- lanes 103-105 – the sentence “For the recruitment, the researchers sent electronic invitations by e-mail to the student of second year of bachelor degree from the Facultad de Deportes Campus Ensenada at Universidad Autónoma de Baja California, in the electronic message, the authors explained briefly the aims of the study”, should maybe be separated into two (suggestion “For the recruitment, the researchers sent electronic invitations by an e-mail to the student of second year of bachelor degree from the Facultad de Deportes Campus Ensenada at Universidad Autónoma de Baja California. In the electronic message, the authors explained briefly the aims of the study”)
- lanes 197-198 – Figure 1 opposite to lane 22 and 99 – control group has the abbreviation CON, but in the graph, CTRL is used for the control group – that should be matched. The same comment applies to 14-minutes (HITT-14) and 21-minutes HIIT (HIIT-21) groups opposite to the abbreviations on the graph. Or in lane 206 the groups are marked in third way. Please, match these abbreviations.
The manuscript doesn’t contain many flaws to be considered for the publishing.
Author Response
The presented article "Online Whole-Body Interval Training Improves Muscle Performance and Spectral Cardiac Autonomic Control in Young Adults" clearly showed that virtual whole-body interval training was an intervention that improved the muscular endurance in young-adults population.
The experimental design is straightforward and the manuscript is pretty well written, although some English proofing would be beneficial. For example:
- lane 4 – there shouldn’t be a period in the headlines
We appreciate your time and knowledge for reviewing this paper. This will help improve our quality of scientific communication.
Changes: We corrected that typo
- lane 40 – there shouldn’t be an “s” in adipose tissue gain,
Changes: We corrected that typo
- lanes 103-105 – the sentence “For the recruitment, the researchers sent electronic invitations by e-mail to the student of second year of bachelor degree from the Facultad de Deportes Campus Ensenada at Universidad Autónoma de Baja California, in the electronic message, the authors explained briefly the aims of the study”, should maybe be separated into two (suggestion “For the recruitment, the researchers sent electronic invitations by an e-mail to the student of second year of bachelor degree from the Facultad de Deportes Campus Ensenada at Universidad Autónoma de Baja California. In the electronic message, the authors explained briefly the aims of the study”)
Response: We thank the reviewer for your suggestion
Changes: We re-structure the sentence with the suggested one
- lanes 197-198 – Figure 1 opposite to lane 22 and 99 – control group has the abbreviation CON, but in the graph, CTRL is used for the control group – that should be matched. The same comment applies to 14-minutes (HITT-14) and 21-minutes HIIT (HIIT-21) groups opposite to the abbreviations on the graph. Or in lane 206 the groups are marked in third way. Please, match these abbreviations.
Response: We thank the reviewer for remarking this flaw in our manuscript
Changes: We corrected the mismatching marks
The manuscript doesn’t contain many flaws to be considered for the publishing.
Reviewer 2 Report
1. The number of subjects is too small compared to the general intervention or prospective experimental study.
2. Although the author stated in the abstract that there is no related study, in the case of a similar study comparing the effects of online cardiorespiratory fitness training using a smartphone, on average, more than 20 participants per group were recruited. Since participants are trained online, which has high accessibility, it will be necessary to recruit more subjects than the existing offline training research.
3. The tool for evaluating training effectiveness is too simplistic. This study tested muscular endurance with push-ups only. However, a commonly used fitness assessment tool is much better for comparing results with previous studies.
4. Training with push-ups and then assessing push-ups is not valid. Is the push-up obviously crucial to the subject's lifestyle?
5. The system description that records data from the app and sends it to the researcher is unclear.
6. With the latest research technology, monitoring the subject's exercise performance status will be necessary. However, the current method is not different from the existing home training.
7. It is necessary to compare basic information of participants by the group.
8. The title is not clear. The content of this study should be explained more directly.
Author Response
- The number of subjects is too small compared to the general intervention or prospective experimental study.
Response: We thank the reviewer for their time and knowledge for revising this paper. This will help improve the quality of our scientific work. Regarding the sample size issue, our a priori calculations with G*Power let us a minimum sample size of 21 for 3 groups and 2 repeated measures, while for the 5 repeated measures of push-ups, the total sample size recommended was 15. The initial participants of our study were 27. However, due to the voluntary withdrawal of some participants, we were left to a final sample of 20. Nevertheless, the total of intervened people was 14 (8 for HIIT-14 and 6 for HIIT-21.
Changes: We added new information regarding the a priori sample size calculation procedure and the reporting sample death.
- Although the author stated in the abstract that there is no related study, in the case of a similar study comparing the effects of online cardiorespiratory fitness training using a smartphone, on average, more than 20 participants per group were recruited. Since participants are trained online, which has high accessibility, it will be necessary to recruit more subjects than the existing offline training research.
Response: We consider sample size as a limitation in the discussion section. Unfortunately, our study underwent a withdrawal from several subjects for personal reasons. This situation reduced the sample size to a marginally required from our a priori analysis.
Changes: Discuss the final sample as a limitation of our study.
- The tool for evaluating training effectiveness is too simplistic. This study tested muscular endurance with push-ups only. However, a commonly used fitness assessment tool is much better for comparing results with previous studies.
Response: Thank you for your observation. The aim of our study was to assess the impact of unsupervised remote training on physical and physiological markers with subjects under lockdown. We, the authors, chose the less intrusive setting intervention possible for the subjects. Therefore, we concluded during the experimental designs that a simple and versatile way to summarize physical fitness was with simple push-ups. Moreover, this assessment is strongly related to cardiovascular markers and physical performance, for more information see below:
Segerström, A.B.; Holmbäck, A.M.; Elzyri, T.; Eriksson, K.-F.; Ringsberg, K.; Groop, L.; Thorsson, O.; Wollmer, P. Upper Body Muscle Strength and Endurance in Relation to Peak Exercise Capacity During Cycling in Healthy Sedentary Male Subjects. J. Strenght Cond. Res. 2011, 25, 1413–1417.
Fennell, C. Effects of Supervised Training Compared to Unsupervised Training on Physical Activity, Muscular Endurance, and Cardiovascular Parameters. MOJ Orthop. Rheumatol. 2016, 5, doi:10.15406/mojor.2016.05.00184.
Changes: We included new information in the push-up test method
- Training with push-ups and then assessing push-ups is not valid. Is the push-up obviously crucial to the subject's lifestyle?
Response: Thanks for your concern regarding the experimental design, to clarify this point, the push up implemented in the HIIT circuit was as many reps as possible for 30 seconds long, combined with other 6 whole body and core exercises, making the push-ups a small fraction of the whole routine (~14%). For the push up test, maximal push-ups until failure is a different setting compared to the HIIT circuit.
According to the scientific evidence, upper-body strength is closely related to overall physical fitness and sport performance (cycling performance).
- The system description that records data from the app and sends it to the researcher is unclear.
Response: Thank you for your comment.
Changes: We re-phrase and detailed that section
- With the latest research technology, monitoring the subject's exercise performance status will be necessary. However, the current method is not different from the existing home training.
Response: Thanks for rising this question in our study. Our aim was to explore the physiological adaptations of interval training during the pandemic enclosure with minimal supervision.
Changes: We clarify our aims by the end of the introduction
- It is necessary to compare basic information of participants by the group.
Response: Thanks for your suggestion.
Changes: We re-arrange the basic information (Table 1) by groups.
- The title is not clear. The content of this study should be explained more directly.
Response: Thank you for signaling that flaw in our manuscript
Changes: We re-adjusted the title to a fitter one.
Reviewer 3 Report
The article has two major problems, one with the choice of the test protocol and the 2nd with the lack of sample size calculation. Please see the attachment for detailed comments.

Author Response
- The vast majority of authors in the field have concluded that interpretations of push-up scores in terms of absolute strength, muscular endurance are invalid, but are moderately valid in terms of muscular strength in relation to body weight.
Response: We greatly appreciate the time and effort from the reviewer to address these concerns and flaws from our study. According to the ACSM, we considered this procedure as a valid test to determine the muscular endurance in the general population. Moreover, upper-body muscle endurance is strongly related to cardiovascular markers and physical performance, for more information see below:
Pescatello, L.; Riebe, D.; Thompson, P. ACSM’s guidelines for exercise testing and prescription; 2014.
Segerström, A.B.; Holmbäck, A.M.; Elzyri, T.; Eriksson, K.-F.; Ringsberg, K.; Groop, L.; Thorsson, O.; Wollmer, P. Upper Body Muscle Strength and Endurance in Relation to Peak Exercise Capacity During Cycling in Healthy Sedentary Male Subjects. J. Strenght Cond. Res. 2011, 25, 1413–1417.
Fennell, C. Effects of Supervised Training Compared to Unsupervised Training on Physical Activity, Muscular Endurance, and Cardiovascular Parameters. MOJ Orthop. Rheumatol. 2016, 5, doi:10.15406/mojor.2016.05.00184.
Changes: We included new information in the push-up test method
- The introduction is very well grounded, but please emphasize the elements of originality/novelty that this study brings to the specialized literature.
Response: We thank the reviewer for their suggestion
Changes: We added the novelty of our work at the end of the introduction.
- The recommendation of an average of 150 minutes per week for the adult population (line 56) lacks bibliographic sources. In a simple search, I found two studies dealing with this issue: Yang, Y. J. An Overview of Current Physical Activity Recommendations in Primary Care. Korean journal of family medicine, 2019, 40 (3), 135 142. https://doi.org/10.4082/kjfm.19.0038
Szabo, D.A. The importance of cardiorespiratory fitness and physical activity among adulthood stages review. Studia UBB Educatio Artis Gymn., 2021, LXVI, 4, pp. 85 101. DOI:10.24193/subbeag.66(4).35
Response: We thank the reviewer for their suggestion
Changes: We added the recommended works to the regular exercise prescription sentence.
- Please change Abril with April (line 126)
Response: Thanks for your suggestion. This will help improve our quality of proofreading.
Changes: We corrected that typo
- I have carefully read the program developed by Schleppenbach et al. (who used The Polar T-31 Transmitter (Polar Electro Oy, Kempele Finland) heart rate monitor was used to measure the participant's heart rate during exercise) and the one modified by you, and I can state that all exercises are used in HIIT type training. However, the problem lies in the choice of the Maximal push-up test, which I repeat, is exactly on the borderline.
Response: We thank the reviewer for their concern. To clarify this issue, the push-up implemented in the HIIT circuit was as many reps as possible for 30 seconds long, combined with other 6 whole body and core exercises, making the push-ups a small fraction of the whole routine (~14%). For the push-up test, maximal push-ups until failure is a different setting compared to the HIIT circuit. Lastly, according to scientific evidence, upper-body strength is closely related to overall physical fitness and sports performance (cycling performance).
- Nowhere in the paper did I find the sample size or statistical power calculation. Please enter the sample size calculation.
Response: We thank the reviewer for their time and knowledge for revising this paper. Regarding the sample size issue, our a priori calculations with G*Power let us a minimum sample size of 21 for 3 groups and 2 repeated measures, while for the 5 repeated measures of push-ups, the total sample size recommended was 15. The initial participants of our study were 27. However, due to the voluntary withdrawal of some participants, we were left to a final sample of 20. Nevertheless, the total of intervened people was 14 (8 for HIIT-14 and 6 for HIIT-21.
Changes: We added new information regarding the a priori sample size calculation procedure and the reporting sample death.
- I recommend that figure number 1 must be introduced into a specialized software to be enhanced.
Response: We thank the reviewer for your suggestion
Changes: We improved the image quality of the figure
- A new suggestion is that the first sentence of the discussion should contain the objectives of the study, so that each result can be developed/interpreted/compared in chronological order.
Response: We appreciate your observation.
Changes: We added the new sentence at the beginning of the discussion section.
- The paper has very low similarity coefficients, but still, one paragraph needs to be reworded entirely: This study helps to demonstrate that remote HIIT is an effective aid to improve strength and cardiac performance during COVID-19 enclosure in collegiate adults.
Response: We thank the reviewer for this comment
Changes: The sentence was reworded and now sounds more harmonic.
Round 2
Reviewer 2 Report
1. It isn't easy to find differences or improvements in the method used in this study from the existing U-, tele-, or mobile-health training methods. COVID-19 increases the value of services in the field, but research has not shown what's creative.
2. Above all, this approach has an advantage over existing offline training because it is easier for more subjects to participate. However, this study has a much smaller sample size than a typical training study. No specific care is required for the participant. Since the approach is not new, now is the time for large-scale or long-term studies.
Author Response
- It isn't easy to find differences or improvements in the method used in this study from the existing U-, tele-, or mobile-health training methods. COVID-19 increases the value of services in the field, but research has not shown what's creative.
Response: We thank the reviewer for their objective observations for the current study. The main purpose is not study the effectiveness of remote interventions per se, but assess the magnitude of its application during forced lockdown. Not looking for a ground-breaking novelty, but report measurable facts during pandemic-induced lockdown.
Changes: We re-structure the aims of the study to clarify this last point.
- Above all, this approach has an advantage over existing offline training because it is easier for more subjects to participate. However, this study has a much smaller sample size than a typical training study. No specific care is required for the participant. Since the approach is not new, now is the time for large-scale or long-term studies.
Response: We thank the reviewer for your comment.
Changes: We added this statement in the conclusion section.